# Mechanical and Durability Assessment of Recycled Waste Plastic (Resin8 & PET) Eco-Aggregate Concrete

Adewumi John Babafemi [1,*], Nina Sirba [1], Suvash Chandra Paul [2] and Md Jihad Miah [3]

1   Department of Civil Engineering, Stellenbosch University, Stellenbosch 7602, South Africa; ninasirba2@gmail.com
2   Department of Civil Engineering, International University of Business Agriculture and Technology, Dhaka 1230, Bangladesh; suvashpl@iubat.edu
3   Division of Architecture and Urban Design, Urban Science Institute, Incheon National University, 119 Academy-ro, Yeonsu-gu, Incheon 22012, Korea; miahmj@inu.ac.kr
*   Correspondence: ajbabafemi@sun.ac.za; Tel.: +27-21-808-4475

**Abstract:** The massive amount of plastic waste in our natural environment is a global concern. In this study, recycling plastic waste to partially replace natural sand in concrete is investigated. The performance of Resin8, a unique combination of all types of plastics and Polyethylene Terephthalate (PET) in concrete, has also been investigated. Replacement contents of 5%, 10%, and 15% for sand by volume were performed. The concrete mixes incorporating recycled plastic waste were tested against a reference concrete mix without plastic. The workability, compressive strength, tensile strength, oxygen permeability index (OPI), and effect of temperature were assessed. Scanning Electron Microscopy (SEM) analysis was conducted on the plastics and plastic concretes, pre- and post-temperature exposure. PET at a replacement content of 10% slightly increased the compressive strength by 2.4%. Regarding the OPI test, all the mixes incorporating recycled plastic waste are classified as "good". When exposed to a temperature of 250 °C, no significant change in compressive strength was observed for the concrete mixes incorporating Resin8 at a replacement content of 15%, and the mixes incorporating PET at a replacement content of 5%, 10%, and 15%. It was clear from the results that both Resin8 and PET are suitable as a partial replacement for sand in concrete.

**Keywords:** recycled plastic; polyethylene terephthalate; compressive strength; tensile strength; oxygen permeability index

## 1. Introduction

The earth is in the midst of an environmental crisis on a scale never seen before. Climate change's ramifications stretch well beyond any previous major global issue. Our planet's survival is contingent upon developing sustainable solutions to address the myriad of issues currently confronting it. As the world's population continues to rise exponentially, one significant environmental challenge is the effective handling of waste, especially plastic waste. Plastic is non-biodegradable, and because it is typically disposed of shortly after use, handling plastic waste continues to be a significant challenge that demands a proactive intervention. Not only is plastic non-biodegradable, but it is also toxic and significantly pollutes the natural area (landfill, ocean) in which it is left. On a global scale, it is reported that of the 14% of packaging plastics (which contributes 50% of global plastic waste) collected from the environment, only 5% is recycled into new plastic, 14% is incinerated, and the remainder winds up on enormous swaths of land, including landfills and the ocean [1]. One study reported that only about 9% of all plastic ever manufactured has been recycled worldwide [2]. Cumulative plastic waste generation, disposal, and projection to 2050 are shown in Figure 1 [1]. Though incinerating waste plastics is a method for recovering energy from plastic waste that is referred to as quaternary recycling [3], this is not a sustainable practice, as it is estimated that the production and incineration of

plastics will emit over 850 million tonnes of greenhouse gases into the atmosphere in 2021, increasing to 2.8 billion tonnes by 2050 [4].

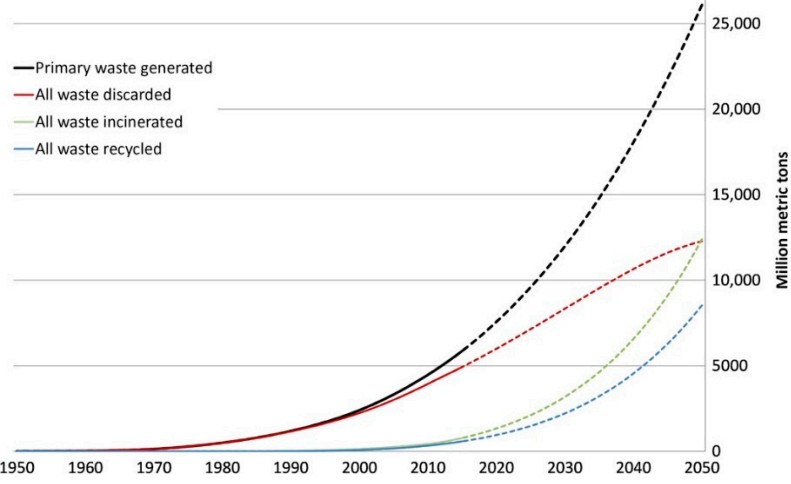

**Figure 1.** Cumulative plastic waste generation and disposal [5].

Recycling plastic waste can be accomplished chemically [6–8] or mechanically [9–11]. Mechanical recycling is now the most common method for recycling post-consumer plastic waste. The waste is ground, washed, and re-melted, for example, via extrusion to create new secondary plastic materials with the same chemical structure as the original [8]. To considerably increase the amount of recycled plastic waste (RPW), it must be performed on a large scale, with the potential for worldwide adoption. Here, the construction industry has an opportunity to contribute significantly to the use of RPW as an eco-aggregate in the production of cement-based materials. Some benefits of the uptake of RPW include employment opportunities for waste collection; the elimination of waste from landfills and the environment; carbon footprint reduction; the building of climate-resilient and energy-efficient infrastructures; the creation of small businesses; sustainability; and the conservation of natural resources [12,13].

It is common knowledge that concrete is the most used material on earth, aside from water. Global consumption is more than 12 billion tonnes per annum [14], with aggregates (fine and coarse) constituting 75–80% of the volume of concrete [15]. The fine aggregate constituent is put at 35–45%. Therefore, even if 5% of RPW is substituted for fine aggregate in concrete, this will eliminate an enormous amount of plastic waste from our oceans, landfills, and neighbourhoods. To this end, several research findings have been published on the use of RPW as an eco-aggregate [16–20] and fibre [2,21–23] in concrete and mortar. One major problem inhibiting the use of RPW in cement composites is the lack of fusion between the plastic aggregate and the cement matrix, resulting in a weak interfacial transition zone (ITZ). The weak ITZ negatively impacts the properties of the RPW cement composites produced. The properties generally reported in the literature on the properties of concrete or mortar containing recycled plastic wastes are the fresh properties (slump), mechanical properties, durability, and microstructural properties [9].

Several authors have reported reduced workability with an increase in the content of RPW when used as sand in concrete [9,24,25]. It should, however, be noted that the plastic's particle size, content, and roughness will influence this property. The air content of RPW concrete is said to increase as the content of RPW increases [9]. Generally, studies have shown that irrespective of the properties of the RPW (type, size, shape, content, etc.), a reduction in the compressive strength is usually obtained when used as a fine aggregate [26–28]. A recent study that used RPW (e-waste plastic) as a partial substitute for natural coarse aggregates also showed that the compressive strength reduces as the RPW content increases [29]. However, the workability increased as the RPW content increased up to 50%. The decreased compressive strength is related to the poor bond between

the recycled plastic aggregate and the cement matrix, resulting in a weaker interfacial transition zone (ITZ) and the lower elastic modulus of the recycled plastic aggregate to natural sand [9,24].

Other mechanical properties of RPW concrete, such as tensile, flexural, and elastic modulus, are also typically less than the control concrete for the same reasons mentioned for the reduction in the compressive strength [9,16,24,28]. However, as obtained with the compressive strength at 10% PET content, an increase (25%) was reported for the tensile strength of the PET-RPW concrete [24]. On the durability performance of RPW used as sand in cement composites, Steyn [9] reported a decrease in the oxygen permeability index (OPI). Still, OPI was classified as good at 15% RPW content. After exposure to an elevated temperature at 20 °C, 600 °C, and 800 °C for 1 h, the mechanical properties (compressive and tensile strength, elastic modulus) reduced at RPW contents of 7.5% and 15% at 600 °C and 800 °C [30]. For electronic plastic waste (EPW) used as a substitute for sand in concrete, abrasion resistance, sorptivity coefficient, ultrasonic pulse velocity (UPV), and alternate wetting and drying tests showed EPW concrete could be used in applications where durability is critical due to satisfactory performance at 10% RPW content. Though mechanical properties are generally lower than control concrete, the merits of RPW concrete will be low density and thermal properties.

In most studies reported in the literature, among the various types of RPW, PET seems to outperform others [13,24]. Furthermore, the optimum content of RPW as a fine aggregate in cement-based materials, as generally reported in the literature, is put at between 5–10% [5]. In this study, a cement-based modifier, Resin8™, an RPW made from all types of mixed and unclean remediated plastic waste (Resin 1–7), has been used to partially replace sand. Resin8 was patented in 2018 by the Centre for Regenerative Design and Collaborations (CRDC), Costa Rica, in partnership with PEDREGAL©. This eco-aggregate could be a breakthrough solution that will have a massive impact on the plastic waste challenge on a global scale. While several researchers have investigated the performance of cement-based composites incorporating different RPW, there seems to be a paucity of study on the performance of an RPW containing all types of resin-based plastic waste (Resin 1–7). The compressive and splitting tensile strength and oxygen permeability index of concrete produced using Resin8 and PET have been investigated and compared to a reference mixture. Furthermore, the compressive strength performance of RPW concrete exposed to a temperature of 250 °C was investigated.

## 2. Materials and Methods

### 2.1. Materials

The materials used in this study are Surebuild Portland cement (CEM II/A-L 42.5N), locally sourced Malmesbury sand and 13 mm Greywacke stone, RPW (Resin8 and PET), and water. RESIN8 is a plastic aggregate that incorporates all 7 types of plastics and has been modified to enhance the performance of cement-based materials. Figure 2 shows the sand, Resin8, and PET aggregates. CRDC, Cape Town, supplied Resin8, while the PET was sourced locally from Supaplas in Belville, South Africa, in processed chips.

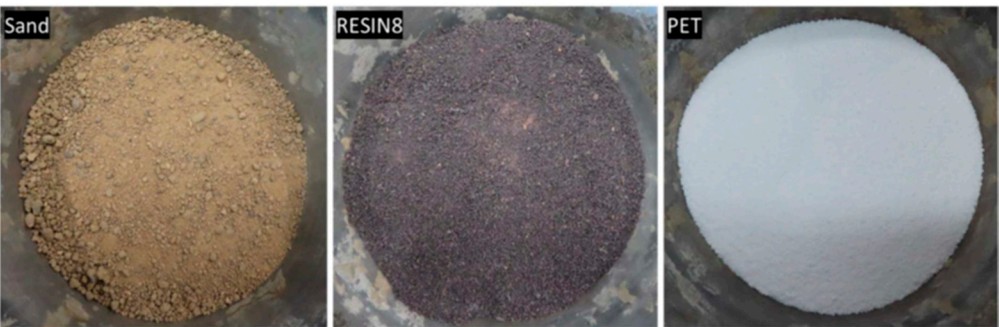

**Figure 2.** Fine aggregates used for mixing.

The sand, Resin8, and PET particle size distributions are presented in Figure 3. The fineness moduli of the sand, Resin8, and PET are 2.74, 2.88, and 2.75, respectively. The research studies of Thorneycroft [24] and Harihanandh & Karthik [25] indicate that RPW is most effective as a fine aggregate in cement-based composites when it has a similar grading to the sand. Therefore, the as-received Resin8 and PET aggregates were milled to have similar grading to the natural sand. As a result, the grading curve in Figure 3 shows that Resin8 and PET have similar curves and are comparable to the sand.

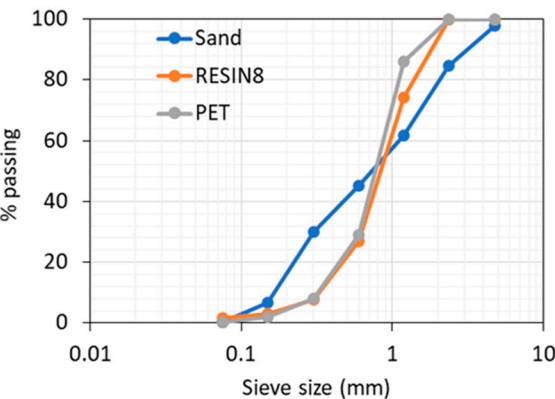

**Figure 3.** Grading curves of sand, Resin8 and PET.

## 2.2. Production of Resin8 RPW

Resin8 is the end product of recycling all mixed and dirty remediated plastic waste (Resin 1–7). Resin8 is an extruded blend of 80% recycled plastic (from various sources) and 20% mineral ingredients (mainly calcium hydroxide and pozzolans). Lime is generally used as an antiseptic filler to remove pollution from plastic.

The production process starts with waste plastic collection (Figure 4a), shredding into flakes (Figure 4b), and the components are combined at temperatures ranging from 190 °C to 200 °C, for low-density polymers, to 230 °C for high-density plastics such as PVC, and extruded into a sausage-like shape (Figure 4c). The production equipment is set to a maximum temperature of 240 °C, as high-density polymers such as PVC produce chlorides or other harmful gases above 260 °C. The extruded environmentally benign mineral-polymer is cooled down in water. Finally, the material is granulated into desired sizes (Figure 4d).

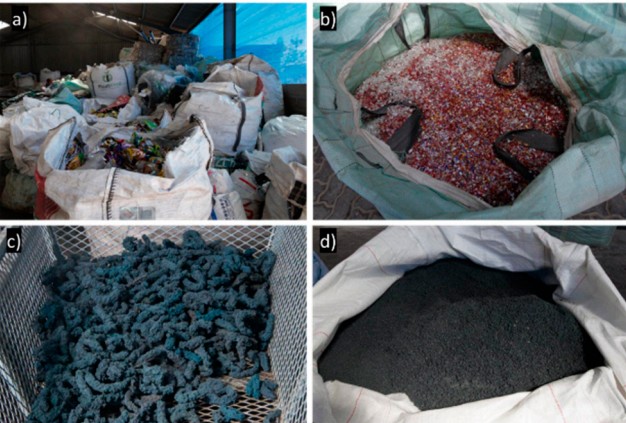

**Figure 4.** Production process for Resin8, (**a**) plastic wastes collected, (**b**) plastic flakes, (**c**) heat extruded plastic before grinding, (**d**) Resin8 plastic aggregate.

### 2.3. Concrete Mix Design

The reference mix (Ref) and those containing Resin8 and PET are shown in Table 1. The RPW were substituted for sand by volume at 5%, 10%, and 15%. The mixes with Resin8 at the various contents are designated as RES5, RES10, and RES15, while those with PET are PET5, PET10, and PET15. It should be noted that the target strength of the reference mix (0% RPW) was 40 MPa with a designed slump of 100 mm. The relative densities of the sand, Resin8, and PET are 2.62, 0.97, and 1.47, respectively.

**Table 1.** Concrete mix design per cubic metre.

| Mix ID | Sand (kg/m$^3$) | Plastic (kg/m$^3$) | Stone (kg/m$^3$) | Cement (kg/m$^3$) | Water (kg/m$^3$) |
|---|---|---|---|---|---|
| Ref | 758 | 0 | 940 | 450 | 225 |
| RES5 | 720 | 14 | 940 | 450 | 225 |
| RES10 | 682 | 28 | 940 | 450 | 225 |
| RES15 | 645 | 42 | 940 | 450 | 225 |
| PET5 | 720 | 21 | 940 | 450 | 225 |
| PET10 | 682 | 42 | 940 | 450 | 225 |
| PET15 | 645 | 63 | 940 | 450 | 225 |

## 3. Experimental Tests

### 3.1. Workability of Mixes

The constituents of the concrete mixes were mixed, and the slump was measured for the 7 mixtures following the procedure prescribed in BS EN 12350-2 [31].

### 3.2. Strength Tests

3.2.1. Compressive Strength

A compressive strength test was conducted for all mixes at 7 and 28 days of curing by complete immersion in water at 23 ± 1 °C. The test was conducted in a KingTest Contest machine at a loading rate of 180 kN/min until failure following BS EN 12390-3 [32]. Test specimens were 100 mm cubes, and the average strength was determined from three replicates.

3.2.2. Splitting Tensile Strength

The splitting tensile test was executed using the Zwick Z250 material testing machine at a loading rate of 0.3 MPa/s. The specimens used were 100 mm concrete cubes and tested at 7 and 28 days. The procedure outlined in BS EN 12390-6 [33] was followed for the test, and three replicates were used per mix and test age.

### 3.3. Durability Tests

3.3.1. Oxygen Permeability Index

Each mixture's oxygen permeability index (OPI) was determined using the procedure outlined in SANS 3001-CO3-2 [34]. This experiment involves pressurising specimens 70 ± 2 mm Ø × 30 mm thick with oxygen and monitoring the pressure drop for 6 h or until the pressure falls below 50 kPa, whichever occurs first. The test was conducted at 35 days of curing the specimens, as it is recommended that the test be conducted after 28 days in the code, and four specimens were used per mix.

3.3.2. Effect of Temperature

Using RPW in concrete raises concerns due to the risk of fire. The effect of temperature on specimens was examined on the concrete's compressive strength after exposure to moderate temperature. The cube is heated from all four sides, which effectively accounts for the worst-case scenario as the concrete is normally only exposed to fire from one direction. Prior to placing the specimens in the oven, they were dried for one day in the temperature-humidity control room. After that, the remaining cubes were baked for 2 h

at 250 °C. The specimens were then slowly cooled in the oven with the door open for another 2 h before being removed and placed in the humidly controlled room for 4 days at a temperature of 22 °C and relative humidity of 50%.

### 3.4. Scanning Electron Microscopy (SEM)

An SEM analysis using the Zeiss MERLIN was performed on the reference mix, mixes with RESIN8 at replacement contents of 5% and 15%, and PET at replacement contents of 5% and 15%. The ITZ and morphology of the mixes, were examined and compared to assess heterogeneity. Energy-dispersive X-ray spectroscopy (EDS) examination was also performed. All specimen images were taken after 28 days.

## 4. Results and Discussion

### 4.1. Workability of Mixes

The workability of the mixes containing the RPW is shown in Figure 5. As shown in Figure 5, the workability increases as the content of the RPW increases, with the PET mixes indicating a higher slump than Resin8. Sand has a higher absorbent property than plastic; hence, higher workability is observed with the RPW concrete. Increases in the workability of concrete as the RPW (e-plastic waste) content increased were also reported by Ullah et al. [28]. The images from a Zeiss MERLIN scanning electron microscope (SEM) presented in Figure 6 indicate that PET has a lower porosity than Resin8, which results in the higher workability of mixes containing PET. Figure 6c,d are higher magnifications of Figure 6a,b. Furthermore, PET particles also have a smoother surface texture and are regular in shape, which enhances the workability of the mix, whereas Resin8 particles have a rough surface texture, which reduces the workability of the mix when compared to PET. Addtionally, due to Resin8's being a combination of all plastic types, it appears more heterogeneous than PET. This could possibly lower the strength properties of Resin8 concrete compared to PET concrete and impact the OPI result.

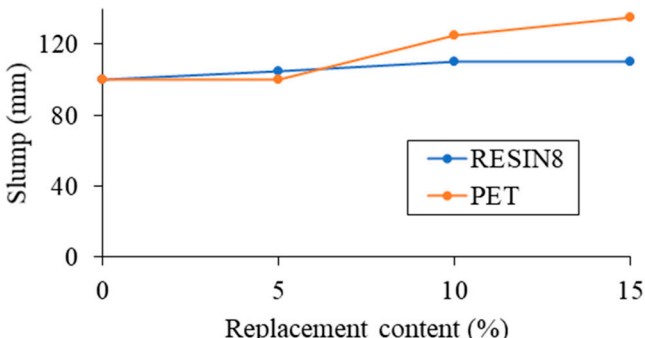

**Figure 5.** Workability of reference and RPW concrete mixes.

### 4.2. Strength Tests

#### 4.2.1. Compressive Strength

The compressive strength of all 7 mixes at 7 and 28 days is shown in Figure 7. Generally, the strength increases with curing age for all mixes (due to the formation of a higher quantity of calcium silicate hydrate (C-S-H) gel for higher curing age) but decreases with an increase in the content of the RPW as expected, especially for Resin8 concretes. In comparison to the reference mix, PET concretes achieved higher strength than the Resin8 concretes at all replacement levels. The PET concrete shows an insignificant strength reduction at 7 and 28 days for 5%, 10%, and 15% PET contents compared to the reference mix. In all mixes, maximum 28-day strength was achieved at 5% of PET content, which is slightly higher than the reference mix by 3%. Furthermore, at 10% PET content, the 28-day strength of the PET concrete was higher than the reference mix by 2.5% but reduced by 4% at PET content of 15%. The slight increase in the compressive strength of PET concrete at both 5% and 10% can be attributed to better aggregate packing and the denser microstructure of PET

compared to Resin8 (Figure 6c,d). According to Thorneycroft [24], when PET with the same grading as the sand was used, increased packing density slightly increased compressive strength (+1.2%) at 10%. However, as the content of PET increases, the strength degrades. This confirms the optimum content of PET reported in the literature [5]. While no strength loss is observed for Resin8 concretes at 28 days between 10% and 15% Resin8 contents, a strength loss of 15% occurs between 5% and 10% Resin8 content or 15% Resin8 content. In comparison to the reference mix, a 28-day strength loss of 4% occurs at 5% Resin8 content and 17% at 10% or 15% Resin8 content.

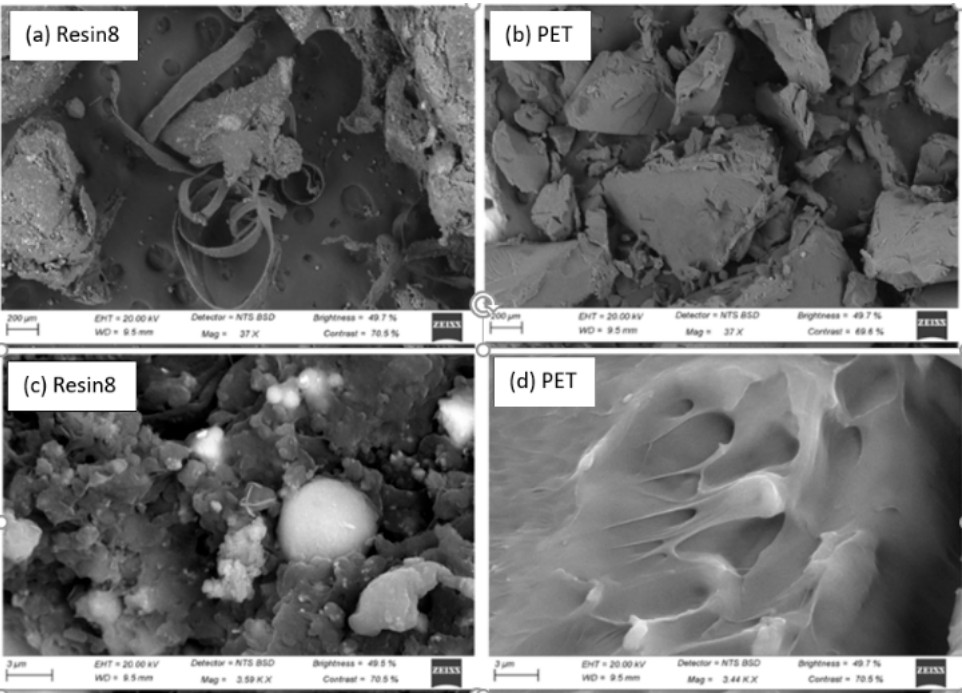

**Figure 6.** Surface morphology of Resin8 and PET at different magnifications (**a**) Resin8, 37× (**b**) PET, 37× (**c**) Resin8, 3.59k× (**d**) PET, 3.44k×.

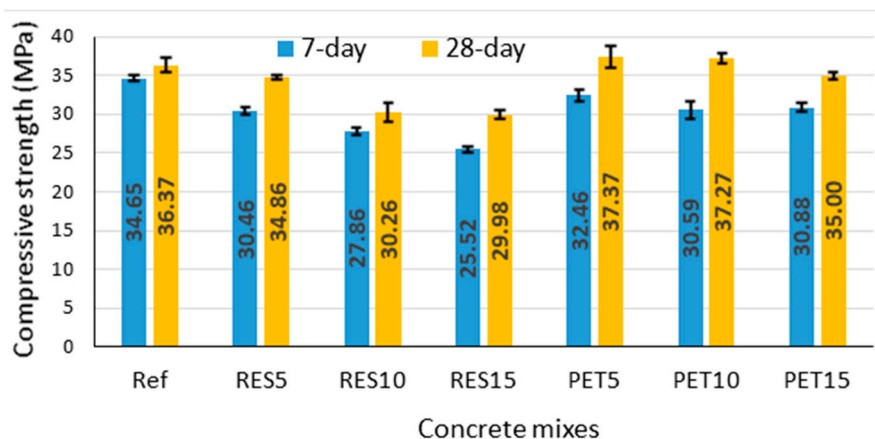

**Figure 7.** Compressive strength of reference and RPW concrete.

The decrease in strength induced by plastics is mostly due to a weak bond between the plastic aggregate and the cement paste, which results in more porous and consequently weaker concrete. This poor bond is generally caused by the smooth surface texture of the plastic [13,16,24]. Additionally, the weak bond is due to the low hydration products formed at the ITZ because of plastic's non-hydrating properties. Another influence on the compressive strength performance is the stiffness of the RPW. Since the RPWs have

lower stiffness and flaky shape compared to natural sand, it is expected that stress concentrations accumulate around them, causing damage and decreasing the mix's compressive strength [35,36].

Furthermore, as seen in Figure 7, PET concrete could have outperformed the Resin8 concrete at all replacement levels because of the porous nature of the Resin8 described in Section 4.1, compressibility under load, and the lower relative density (RD). The RD of fine aggregate influences the strength of the concrete mix [16]. The RD of the RPW concretes also influenced the density of the mixes. Because the RD of Resin8 is lower than PET, its incorporation into concrete results in a higher loss in density. Compared to the reference concrete, the decrease in density is 2.2%, 3.3%, and 4.9% at RES5, RES10, and RES15, respectively. It is 0.9%, 1.69%, and 4% at PET5, PET10, and PET15, respectively. Overall, depending on the required performance, PET15 could be substituted for sand (28-day strength reduction of 4% compared to the reference) and RES5 (strength reduction of 4.3%).

In general, the compressive strength of the plastic concrete compares to the control, with the Resin8 concrete showing the least strength at 10% and 15%. Therefore, concrete containing waste plastic may be used partially as an alternative to ordinary concrete. However, in the development of concrete containing RPW, it is important to suppress the decrease in the concrete strength, and improving the strength is an issue for future research.

### 4.2.2. Splitting Tensile Strength

Figure 8 shows the splitting tensile strength performance of the reference and RPW concretes at 7 and 28 days. Similar to the compressive strength performance, the tensile strength decreases as the content of the RPW increases in the mixtures for both PET and Resin8. The 28-day tensile strength reduction for RES5, RES10, and RES15 is 12.5%, 14%, and 19%, respectively, compared to reference concrete. Similarly, PET5, PET10, and PET15 reduced in tensile strength by 5%, 9.5%, and 11%, respectively, compared to Ref at 28 days. It can be seen from Figure 8 that PET concretes show higher performance than the Resin8 concretes, as seen for the compressive strength performance. When the splitting tensile strength development was evaluated from 7 to 28 days, it was observed that the average rise in strength of all the mixes was 14.2%, which was greater than the compressive strength increase of 11.9 %.

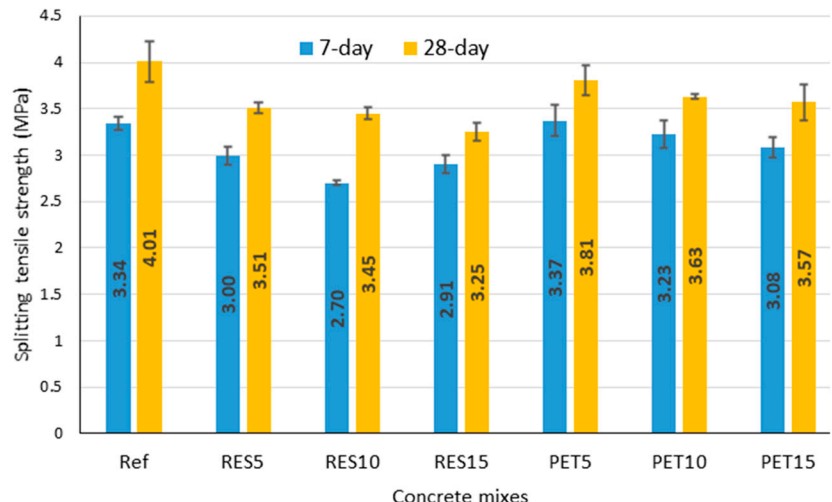

**Figure 8.** Splitting tensile strength of reference mix and RPW concretes.

Sand particles with a rough surface texture and a high stiffness add to the mix's increased tensile strength as with the compressive strength. Because RPW particles are irregular in shape, flaky, and less stiff than the sand particles, the splitting tensile strength of the mix is reduced [13]. Additionally, the weak link between the plastic and the ce-

ment paste leads to the loss of splitting tensile strength, as has been reported by other researchers [9,16]. Furthermore, the lower performance of the Resin8 mixture is due to its being slightly larger than the PET particles. Hence, more voids are created and possibly collect free water, thereby resulting in reduced performance.

In addition, the concrete containing plastic aggregate could undergo higher shrinkage due to the lower strength, stiffness, and flaky shape of plastic aggregate compared to the other constituents of concrete (i.e., sand and stone aggregates) [13]. This results in higher shrinkage cracks in the concrete matrix (i.e., the weak bond among the particles), thus compromising the tensile strength of the concrete.

However, the RPW mixtures show more ductility (no brittle splitting failure) than the reference mix, as shown in Figure 9. This is due to the flexibility of the RPW granules acting as a bridge between micro-cracks before attaining ultimate strength. This behavior correlates with those reported by other researchers [9,37].

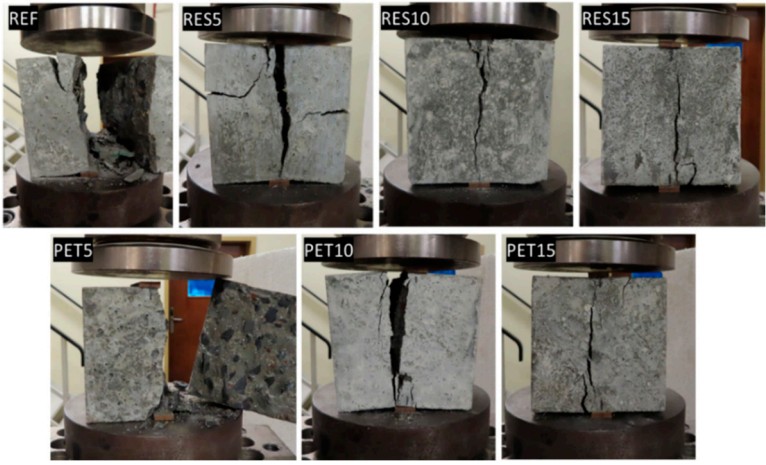

**Figure 9.** Split tensile failure mode of specimens.

### 4.2.3. Relationship between Compressive and Splitting Tensile Strength

Figure 10 illustrates the percentage change in compressive and splitting tensile strength for all plastic-containing mixes compared to the reference mix. The percentage change in compressive and splitting tensile strength for Resin8 is demonstrated to be fairly similar. More specifically, there is a greater decrease in splitting tensile strength for PET-concrete than there is a decrease in compressive strength. As shown in Figure 10, the concrete containing 5% and 10% PET aggregate enhanced the compressive strength rather than declined. In contrast, all the mixes containing Resin8 decreased both compressive and tensile strength, and the reductions were similar (except for the mix with 10% Resin8). These results conspicuously showed that the mechanical strength properties of the concrete mix with the PET aggregate performed better than with the Resin8 aggregate.

### 4.3. Durability Tests—Oxygen Permeability Index (OPI)

Concrete permeability refers to its ability to transport gas or liquid by permeation, which depends on the microstructure and the moisture content in the materials. The OPI performance of all seven mixes is presented in Table 2. Because the correlation coefficient ($r^2$) for all specimens is greater than 0.99, all results were determined to be valid. Typically, gas permeability through a specimen, which indicates the connectedness of pores, should be in the range of 8–11 [38]. Higher values indicate decreased gas permeability and hence improved durability. From the results, it is evident that the PET-concrete at all replacement levels (PET5, PET10, PET15) performed better than the Resin8-mixtures having higher OPI. These results agreed with the compressive and tensile strength of the same mix, as shown in Figures 7 and 8. Indeed, cracks and their orientation significantly affect the permeability of concrete, i.e., the higher the cracks, the higher the permeability [39,40], and the lower

the strength. Compared to the reference, PET5 shows better performance, while PET10 is comparable to the reference mix. RES5, RES10, and RES15 reduce the OPI by 1.9%, 3.1%, and 3.7%, respectively, more than the reference mix. This finding agreed with the lower mechanical strength of the same mix, as shown in Figures 8 and 10. PET5 increases the OPI by 1.2%, whereas PET10 and PET15 decrease the OPI by 0.1% and 0.9%, respectively. The results obtained in this study show more improved OPI than for the plastic concrete reported in Steyn et al. [9].

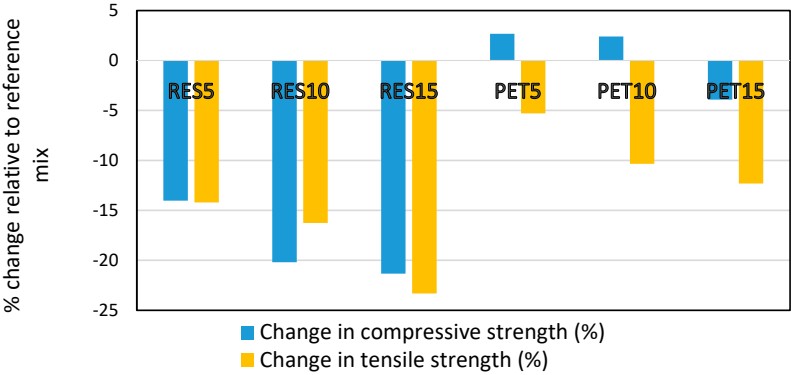

**Figure 10.** Per cent change in compressive and tensile strength of RPW concrete compared to the reference.

**Table 2.** OPI test results of reference and RPW concrete.

| Mixes | OPI (log Scale) | k ($10^{11}$) m/s |
|---|---|---|
| Ref | 10.36 | 4.81 |
| RES5 | 10.16 | 7.06 |
| RES10 | 10.04 | 9.11 |
| RES15 | 9.98 | 11.10 |
| PET5 | 10.48 | 3.33 |
| PET10 | 10.35 | 5.17 |
| PET15 | 10.27 | 5.35 |

Resin8 granules have previously been shown to be more porous than their PET counterpart (as shown in Figure 4) and slightly larger in particle size. Hence, the OPI results are better for PET-concrete than Resin-concrete. The higher porosity and particle size of Resin8 create an interlink between pores, thus leading to a decrease in the OPI [9]. Additionally, the gap at the ITZ between the plastic and cement paste can also influence the OPI. As discussed in Section 4.5, PET-concrete shows a lesser gap at the ITZ than Resin8-concrete, which may directly affect the permeability of the concrete.

### 4.4. Effect of Temperature on Compressive Strength

After exposing the samples to an electric furnace at 250 °C for 2 h, no significant change in compressive strength was observed, as evident in the results presented in Table 3.

**Table 3.** Pre- and post-temperature exposure effect on compressive strength.

| Mixes | Compressive Strength (MPa) | |
|---|---|---|
| | Pre-Temperature | Post-Temperature |
| RES15 | 29.98 | 28.87 |
| PET5 | 37.37 | 37.94 |
| PET10 | 37.27 | 37.16 |
| PET15 | 35.00 | 35.62 |

Generally, the PET aggregate melts at a lower temperature range from 250–275 °C [41,42]. It is well known that a PET aggregate melts during heating and hardens throughout cooling. Therefore, both studied plastic aggregates were melted while heating the concrete cubes

and sticking them to the coarse aggregate and cement paste. It should be noted that after heating the specimens with plastic aggregates (Resin8 and PET), they were left inside the humidly controlled room for 4 days at a temperature of 22 °C and with relative humidity of 50%, which allowed the acceleration of the bonding among the melted plastic aggregates, stone aggregate, and cement paste. Furthermore, as the specimens were stored for 4 days with 50% humidity, new portlandite and C-S-H gels may form due to the rehydration of dehydrated products [43] by absorbing the moisture. This may heal the microcracks and fill the large capillary pores, thus reducing the permeability and porosity of concrete. The process would be more accelerated with the presence of an unhydrated binder and the quantity in the specimens. This combined action may compensate for the possible strength reduction of concrete specimens with plastic aggregates induced by the exposure at 250 °C. However, as the strength reduction due to high temperature is almost negligible, these results and mechanical strength test results without thermal exposure (as reported in Figures 7 and 8) hinted that both Resin8 and PET are suitable as a partial replacement for sand in concrete.

*4.5. SEM Analysis*

The SEM images of the surface morphology of the reference mix, RES5, RES15, PET5, and PET15, are shown in Figure 11. A comparison of the reference mix and the plastic mixes show that plastic mixes have more pores (air voids) than the reference mix, with Resin8-concrete showing more than PET-concrete. The particle shape of Resin8-concrete also influences the level of air voids [13]. Moreover, higher heterogeneity is seen with the plastic mixes than with the reference. A more heterogeneous mixture introduces non-uniform stresses and strains, resulting in stress concentrations that tend to be explicitly concentrated at the ITZ, reducing the mix's strength [44]. This confirms the reason for the lesser strength performance of Resin8-concrete than PET-concrete.

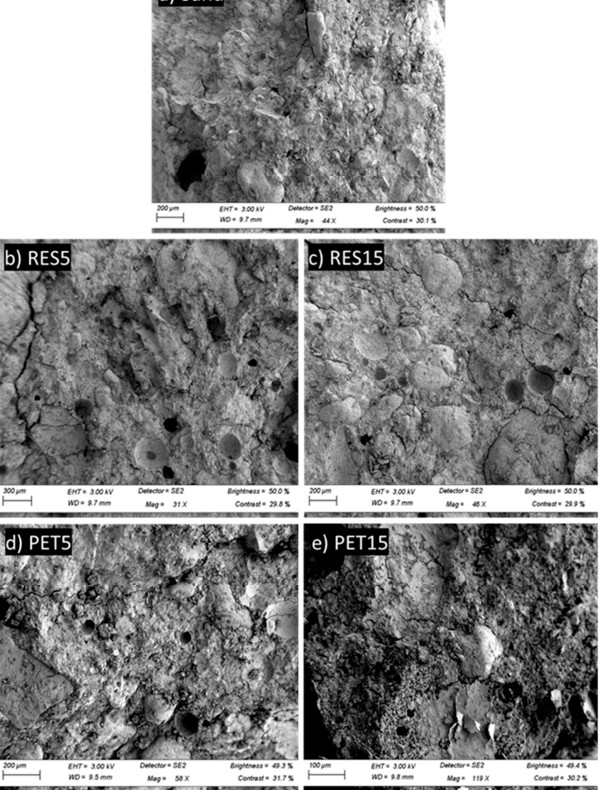

**Figure 11.** Surface morphology of (**a**) reference; (**b**) RES5 (**c**) RES15 (**d**) PET5, and (**e**) PET15.

A closer look at the plastic aggregate-cement matrix ITZ reveals a weaker bond than the reference concrete, which is evident in the visible gap formed (Figure 12a–e). As previously discussed, this impacted both the mechanical and durability properties of the RPW concrete. In addition, due to the less hydrophobic nature of plastics, extra surface water is retained at this gap, thereby resulting in lesser C-S-H (calcium silicate hydrate) and CH (calcium hydroxide) crystals, a porous structure, and weaker aggregate–matrix bonds [5,45]. However, an improvement in the bond between the plastic aggregate and cement paste can be achieved by surface treatment of the plastic using oxidising agents such as a hydrogen peroxide solution (H$_2$O$_2$) and calcium hypochlorite solution (Ca(ClO)$_2$) [5,46]. Generally, plastic aggregates treated with these oxidising agents create chemical species on the surface of the aggregates, which can improve the bonding between the cement paste and aggregates and lead to higher strengths.

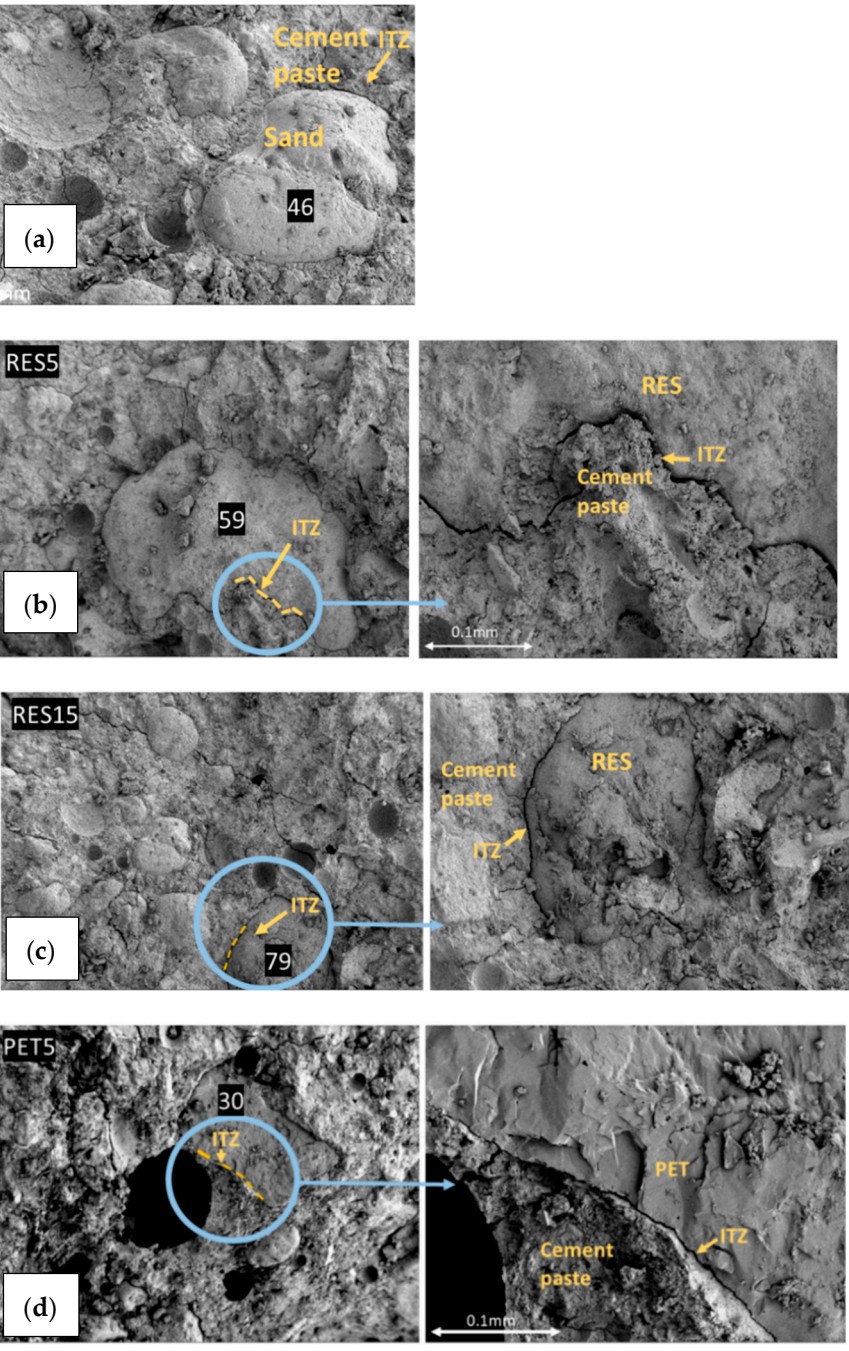

**Figure 12.** *Cont.*

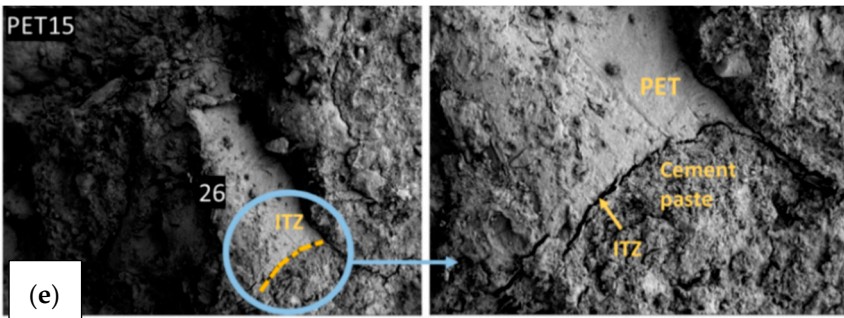

**Figure 12.** ITZ between aggregate and cement paste for all mixes (**a**) Reference (**b**) RES5 (**c**) RES15 (**d**) PET5 (**e**) PET15.

## 5. Conclusions

In this experimental investigation, the compressive strength, splitting tensile strength, oxygen permeability index, and post-fire behavior of concrete containing recycled plastic waste as an eco-aggregate for sand replacement at 5%, 10%, and 15% have been studied. From the outcomes of this study, the following conclusions can be made:

1. Increasing the plastic content in the mix improves the workability of the concrete. Resin8 was less workable than PET, as the PET concrete slumped gradually, whereas the Resin8 concrete declined rapidly.
2. A decrease in density and compressive strength were observed generally. Compressive strength increased with PET replacement content of 5% and 10% but decreased with PET replacement content of 15%. PET at a 10% replacement content increased the compressive strength of the mix by 2.5%, while Resin8 at a 5% replacement content decreased it by 4%.
3. Splitting strength for all mixes decreased as the plastic content increased. PET concrete showed better strength at 10% and 15% than Resin8 at 5%.
4. The plastic concrete's OPI results indicated that the optimum replacement content for Resin8 and PET was 15%, indicating that all mixes are "good". PET increased the OPI by 1.5% more than the reference mix.
5. An almost negligible reduction in compressive strength of concrete containing Resin8 and PET after exposure to 250 °C has been observed. This behavior could be attributed to the healing of microcracks and filling of the large capillary pores (i.e., lowering the permeability and porosity) induced by sticking melted plastic to the coarse aggregate and cement paste and the possible formation of new portlandite and C-S-H gels caused by the rehydration of dehydrated products.

This study has shown that PET and Resin8 have the potential to be used as substitutes for natural sand in concrete. However, whether or not recycled plastic waste can be used as a structural material where high strength is required is an issue for future study. Moreover, the bond strength of embedded reinforcement in plastic concrete and corrosion resistance require investigation.

**Author Contributions:** Conceptualisation, methodology, project administration, supervision, writing—original draft preparation, A.J.B., Conceptualisation, methodology, investigation, formal analysis, visualisation, N.S., writing—review and editing, S.C.P. and M.J.M. All authors have read and agreed to the published version of the manuscript.

**Funding:** This research received no external funding.

**Institutional Review Board Statement:** Not applicable.

**Informed Consent Statement:** Not applicable.

**Data Availability Statement:** The data presented in this study are available upon request from the corresponding author.

**Acknowledgments:** The first author acknowledges the Department of Civil Engineering, Stellenbosch University, for access to facilities to conduct the tests and the provision of materials.

**Conflicts of Interest:** The authors declare no conflict of interest.

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
