# Peer review of "Mechanical and Durability Assessment of Recycled Waste Plastic (Resin8 & PET) Eco-Aggregate Concrete"

_sustainability, doi:10.3390/su14095725_

Round 1

Reviewer 1 Report

In this manuscript, the authors performed mechanical and durability assessment of recycled waste plastic (Resin8 & PET) as eco-aggregate concrete. Before it can be considered for publication, the following revisions are needed.

Abstract:

  1. Good and clear.

Introduction:

  1. Page 3, Line 89-91. ‘After exposure to an elevated temperature at 200 °C, 600 °C, and 800 °C for 1 hr, the mechanical properties (compressive and tensile strength, elastic modulus) reduced at RPW contents of 7.5% and 15% [28].’ Authors should explain the 7.5% and 15% is for 200 °C and 600 °C

Methodology:

  1. Section 2.1: Line 115-116. Authors stated that ‘RESIN8 is a plastic aggregate that incorporates all 7 types of plastics and has been modified to enhance the performance of cement-based materials’, does it means that PET falls under RESIN8?

Results and Discussion:

  1. Overall clear and good work.

Author Response

Dear reviewer,

Thank you for your time to review the article and provide valuable comments to improve the article. The comments have been taken into account and the responses to each comment can be found in the attachment.

Reviewer 2 Report

Comments

In this paper,the following improvements are desired .

  1. Please describe the chemical composition, softening point, and hardness of Resin-8. Then, I would like you to discuss the strength and heat resistance of concrete composite materials based on those data,

  1. I think that the aggregate has the effect of suppressing the shrinkage of the cement paste. Please describe the difference in the amount of shrinkage of each composite when sand, Resin-8, and PET are mixed with cement. Furthermore, I would like you to discuss the strength of the composite based on the amount of shrinkage.

  1. Certainly, by mixing waste plastic with cement, waste plastic can be reduced. Also, in recent years, there is a shortage of natural sand and stones. It would be great if waste plastic could be used as an aggregate for concrete. However, when waste plastic is used as an aggregate, there is a demerit that the strength of concrete decreases. The benefits of using waste plastic as an aggregate should be shown more clearly.

Author Response

Dear Reviewer,

The authors appreciate your time in reviewing the manuscript and comments made to improve the manuscript. We have now taken all your comments into account in improving the manuscript. Responses to all your comments can be found in the attachment. Also, the revised manuscript is attached for your attention.

Round 2

Reviewer 2 Report

Comments

When waste plastic is used as the aggregate, the strength of concrete decreases. In this paper, the primary factor or mechanism of the decrease in strength is not clarified. Even if concrete containing waste plastic meets the strength criteria, it is too early to say that it can be used as an alternative to ordinary sand concrete because the main cause of the decrease in strength has not been clarified.

The strength of this concrete is not very high, and its quality and corrosion resistance are unknown. If the concrete is used in structures such as bridges and buildings that require high strength and corrosion resistance, some problems may occur in the future. Safety concerns remain.

I think that author should just say, "Concrete containing waste plastic may be able to be used partially as an alternative to ordinary concrete." Furthermore, I would like you to write clearly that whether or not it can be used for structural materials that require high strength is an issue for future study. 

I would like you to write clearly that "in the development of concrete containing waste plastic, it is important to suppress the decrease in the concrete strength, and improving the strength is an issue for future research”.

Author Response

The authors would like to appreciate the reviewer again for his constructive comments meant to improve the manuscript. The authors have implemented further changes to the manuscript based on the reviewer's additional comments.

(1) The grammar has been revised further in the areas identified by the authors.
(2) Additional statements have been added to the introduction between lines 85-87.
(3) Asides from the mechanisms discussed for the suppressed strength in Section 4.2.1, additional comments have now been added between lines 234 to 239, and 265 to 270. These include the specific statements that the reviewer recommended as an addition to the manuscript.
(4) Additional statements, including recommended statements by the reviewer have also been added from lines 428 to 432.

Round 3

Reviewer 2 Report

The manuscript has been revised well.